# EEG-fMRI in awake rat and whole-brain simulations show decreased brain responsiveness to sensory stimulations during absence seizures

Petteri Stenroos[1,2], Isabelle Guillemain[1], Federico Tesler[3], Olivier Montigon[1,4], Nora Collomb[4], Vasile Stupar[1,4], Alain Destexhe[3], Veronique Coizet[1], Olivier David[1,5†], Emmanuel L Barbier[1,4*†]

[1]University Grenoble Alpes, Inserm, U1216, Grenoble Institut Neurosciences, Grenoble, France; [2]A.I. Virtanen Institute for Molecular Sciences, University of Eastern Finland, Kuopio, Finland; [3]Paris-Saclay University, CNRS, Institut des Neurosciences (NeuroPSI), France, Saclay, France; [4]University Grenoble Alpes, Inserm, US17, CNRS, UAR 3552, CHU Grenoble Alpes, IRMaGe, Grenoble, France; [5]Aix Marseille University, INSERM, INS, Inst Neurosci Syst, Marseille, France

**\*For correspondence:** emmanuel.barbier@univ-grenoble-alpes.fr

†These authors contributed equally to this work

**Competing interest:** The authors declare that no competing interests exist.

**Sent for Review** 11 July 2023
**Preprint posted** 28 July 2023
**Reviewed preprint posted** 16 October 2023
**Reviewed preprint revised** 13 February 2024
**Reviewed preprint revised** 30 May 2024
**Version of Record published** 08 July 2024

**Abstract** In patients suffering absence epilepsy, recurring seizures can significantly decrease their quality of life and lead to yet untreatable comorbidities. Absence seizures are characterized by spike-and-wave discharges on the electroencephalogram associated with a transient alteration of consciousness. However, it is still unknown how the brain responds to external stimuli during and outside of seizures. This study aimed to investigate responsiveness to visual and somatosensory stimulation in Genetic Absence Epilepsy Rats from Strasbourg (GAERS), a well-established rat model for absence epilepsy. Animals were imaged under non-curarized awake state using a quiet, zero echo time, functional magnetic resonance imaging (fMRI) sequence. Sensory stimulations were applied during interictal and ictal periods. Whole-brain hemodynamic responses were compared between these two states. Additionally, a mean-field simulation model was used to explain the changes of neural responsiveness to visual stimulation between states. During a seizure, whole-brain responses to both sensory stimulations were suppressed and spatially hindered. In the cortex, hemodynamic responses were negatively polarized during seizures, despite the application of a stimulus. The mean-field simulation revealed restricted propagation of activity due to stimulation and agreed well with fMRI findings. Results suggest that sensory processing is hindered or even suppressed by the occurrence of an absence seizure, potentially contributing to decreased responsiveness during this absence epileptic process.

## eLife assessment

This study conducted fMRI experiments in an inbred rat model of absence seizures. The results provide new information suggesting reduced brain responsiveness during this type of seizure. The reviewers had divergent opinions but on average thought the study was **valuable** and the conclusions were **solid**.

## Introduction

Absence seizures typically appear in children between the ages of 5 and 7, and are characterized by a sudden, brief impairment of consciousness, an interruption from ongoing activities, and an unresponsiveness to environmental stimuli (*Loiseau et al., 1995*). These seizures typically last for a few seconds

to several minutes and include a regular electroencephalographic pattern known as spike-and-wave discharge (SWD) with a frequency of 2–5 Hz (*Szaflarski et al., 2010*). In non-treated patients, absence seizures can occur from a few to hundreds of times per day (*Loiseau et al., 1995*). The impaired consciousness during absence seizures can be highly disabling. Indeed, absence epilepsy is not a benign condition and is often accompanied by severe neuropsychiatric comorbidities including impairment of attention, memory, and mood (*Crunelli et al., 2020*). To better understand how these comorbidities arise, it is important to investigate how information processing is altered between the ictal and interictal periods.

In absence epilepsy, human neuroimaging studies suggest that during a seizure, there may be a lack of conscious information processing due to impaired frontoparietal network, arousal systems in the thalamus and brainstem (*Blumenfeld, 2012*), or default mode network (*Luo et al., 2011*). Likewise, multiple human studies on absence seizure have demonstrated lack of responsiveness to external stimuli such as commands and questions, potentially caused by focal disruption of information processing in specific corticothalamic networks (*Blumenfeld, 2012*). However, to our knowledge, current neuroimaging studies on absence epilepsy have been conducted in a resting state, without external stimulus. Therefore, it remains unclear how the brain manages environmental stimuli during absence discharges, and more research is needed to understand how the brain responsiveness is affected during altered brain states.

The GAERS is a well-established model of absence epilepsy. It is based on the selection of rats exhibiting spontaneous SWD and recapitulates most electrophysiological, behavioral, and pharmacological features of human absence epilepsy (*Depaulis et al., 2016*). Although the origin of absence seizures is not fully understood, current studies on rat models of absence seizures suggest that they arise from excitatory drive in the barrel field of the somatosensory cortex (*David et al., 2008*; *Meeren et al., 2002*; *Polack et al., 2007*) and then propagate to other structures (*David et al., 2008*) including thalamus, knowing to play an essential role during the ictal state (*Huguenard, 2019*). Notably, thalamic subnetwork is believed to play a role in coordinating and spacing of SWDs via feed-forward inhibition together with burst firing patterns. This leads to the rhythms of neuronal silence and activation periods that are detected in SWD waves and spikes (*Huguenard, 2019*; *McCafferty et al., 2023*).

Previous results on GAERS have indicated that, during an absence seizure, hyperactive electrophysiological activity in the somatosensory cortex can contribute to bilateral and regular SWD firing patterns in most parts of the cortex. These patterns propagate to different cortical areas (retrosplenial, visual, motor, and secondary sensory), basal ganglia, cerebellum, substantia nigra, and thalamus (*David et al., 2008*; *Polack et al., 2007*). Although SWDs are initially triggered by hyperactive somatosensory cortical neurons, neuronal firing rates, especially in majority of frontoparietal cortical and thalamocortical relay neurons, are decreased rather than increased during SWD, resulting in an overall decrease in activity in these neuronal populations (*McCafferty et al., 2023*). Previous functional magnetic resonance imaging (fMRI) studies have demonstrated blood volume or BOLD signal decreases in several cortical regions including parietal and occipital cortex, but also, quite surprisingly, increases in subcortical regions such as thalamus, medulla, and pons (*David et al., 2008*; *McCafferty et al., 2023*). In line with these findings, graph-based analyses have shown an increased segregation of cortical networks from the rest of the brain (*Wachsmuth et al., 2021*). Altogether, alterations in these focal networks in the animal models of epilepsy impair cognitive capabilities needed to process specific concurrent stimuli during SWD and therefore could contribute to the lack of behavioral responsiveness (*Chipaux et al., 2013*; *Luo et al., 2011*; *Meeren et al., 2002*; *Studer et al., 2019*), although partial voluntary control in certain stimulation schemes can be still present (*Taylor et al., 2017*).

The objective of this study was to investigate changes in whole-brain responsiveness to sensory stimuli during ictal and interictal states, using the GAERS animal model. To avoid the potential impact of anesthetic agents on fMRI recordings and because absence seizures can be only observed in awake animals, EEG-fMRI was performed in GAERS trained to remain still and awake, i.e., non-curarized and non-anesthetized, using a previously validated protocol (*Paasonen et al., 2018*; *Stenroos et al., 2018*). For this purpose, the functionality of the zero echo time (ZTE) sequence was first piloted, and selected over traditional echo-planar imaging (EPI) sequence for its lower acoustic noise and reduced magnetic susceptibility artifacts. The selected MRI sequence thus appeared optimal for awake EEG-fMRI measurements. Visual and somatosensory whisker stimulations were used during both ictal and

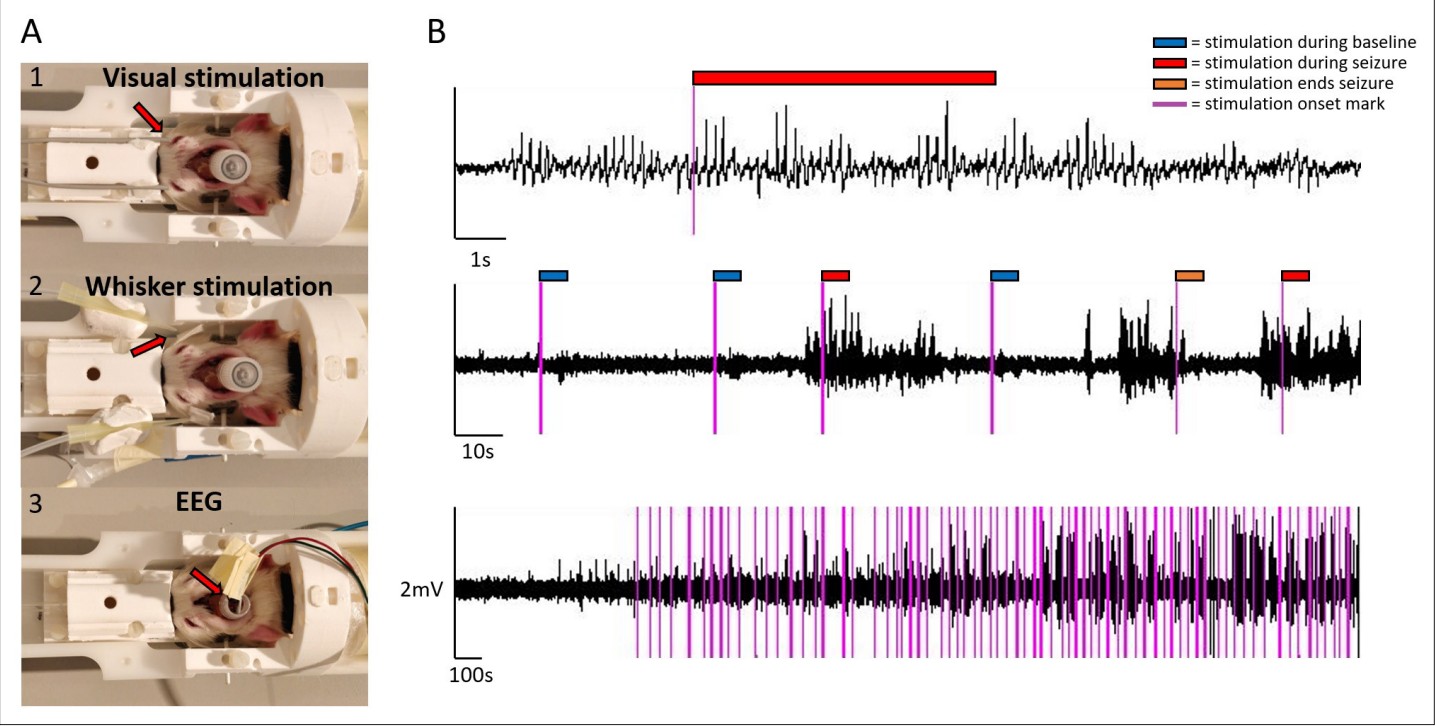

**Figure 1.** Sensory stimulation, EEG setup, and example EEG traces. For visual stimulation, optical fiber cables were positioned bilaterally close to the eyes (**A1**) .For whisker stimulation, plastic tips guided air flow bilaterally to whiskers (**A2**). For EEG, carbon fiber leads were connected to electrodes coming from a plastic tube (**A3**). EEG traces are illustrated at three different temporal scale, during ictal and interictal states and with stimulation onset marks (**B**). Color tags mark stimulation onsets (purple) and the 6 s stimulation blocks during baseline (blue), seizure (red), and when stimulation ended a seizure (orange). The onset marks were added post hoc, based on recorded TTL events.

The online version of this article includes the following figure supplement(s) for figure 1:

**Figure supplement 1.** Example of EEG trace and illustration of stimulation and seizure block paradigms used as statistical parametric mapping (SPM) inputs.

**Figure supplement 2.** Habituation and imaging schedules for three illustrative rats.

interictal states, as previous research has shown altered electrophysiological activity and behavior in these sensory systems during seizures (*Meeren et al., 2002*; *Meyer et al., 2018*; *Pavone et al., 2001*; *Studer et al., 2019*). By using both stimulation schemes, we aimed to investigate alterations in brain responsiveness in each sensory system and to identify any common changes. To further describe the change when switching between ictal and interictal states, the whole-brain response to visual stimulus in each state was simulated, using a recent mean-field model (*Volo et al., 2019*).

## Results

### Fine-tuning of fMRI recording in awake rats

Using our EEG-fMRI setup with transmit-receive loop coil (*Figure 1*) the spatial signal-to-noise ratio (SNR) was ~25 whereas the temporal SNR was between 30 and 60 in the brain (*Figure 2*). In pilot studies, responses to visual stimulation measured using ZTE and EPI sequences were compared in one animal (*Figure 2—figure supplement 1*). Average signal change due to stimulation was between 1% and 1.5% with gradient-echo (GE)-EPI sequence while it was ~0.3% with ZTE. However, activation maps conducted with Aedes, https://github.com/mjnissi/aedes, copy archived at *mjnissi, 2024* demonstrated less susceptibility-induced image distortions in ZTE sequence compared to EPI. Moreover, when studying the effect of fMRI artifacts on EEG signal, more pronounced and higher amplitude gradient switching artifacts were detected when using EPI compared to ZTE sequence.

When comparing peak acoustic noise levels inside the 9.4 T Bruker magnet, ZTE (78.7 dB) was 35.8 dB quieter than EPI (114.5 dB) equaling ~62 times weaker sound pressure (*Schomer, 1998*). Motion

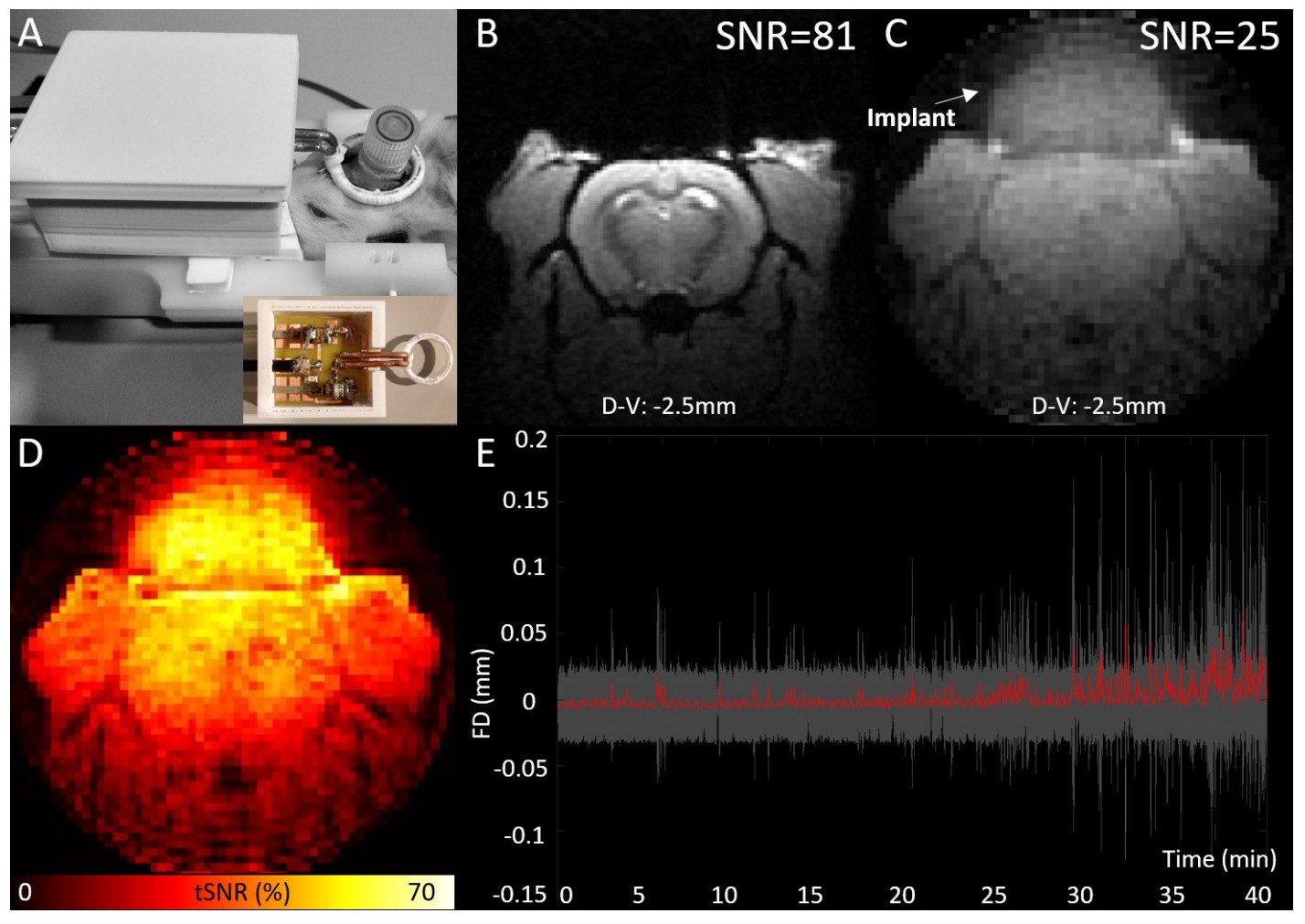

**Figure 2.** EEG-functional magnetic resonance imaging (fMRI) setup and illustration of MRI data quality. MRI transmit-receive loop coil placed around the implant (**A**), spatial signal-to-noise ratios of an illustrative high-resolution T1-FLASH (**B**), and low-resolution zero echo time (ZTE) image (**C**), temporal signal-to-noise ratios (tSNRs) of ZTE data (**D**) from a one example animal and average framewise displacement (red) with the standard deviation (gray area) across all sessions included to analyses (**E**).

The online version of this article includes the following figure supplement(s) for figure 2:

**Figure supplement 1.** Zero echo time (ZTE) and gradient-echo-echo-planar imaging (GE-EPI) data collected during visual stimulus experiment (15s ON, 30s OFF paradigm), and EEG trace illustrating MRI gradient artifacts, from an example rat in preliminary study.

during fMRI using ZTE was low (*Figure 2B*). Mean framewise head translation was only 0.69±0.32 μm and head rotation 0.79±0.55°. Maximum displacement of the rat head was 12.8±11.6 μm, corresponding to 0.03±0.02 voxels, while maximum rotation was 18.1±12.9°. Motion occurrences between the current study (0.43±0.45 motions/min) and the previous study (1.0±0.20 motions/min), performed with a similar rat restraint holder but an EPI sequence (*Stenroos et al., 2018*), demonstrated the advantage of using the quieter ZTE sequence compared to the louder EPI sequence. ZTE motion levels were similar to that obtained in a previous study using a quiet MB-SWIFT sequence (0.48±0.23 motions/min, *Paasonen et al., 2020*).

In our study, use of 3 out of 11 animals (27%) had to be discontinued due to noisy EEG signal, most likely caused by partly detached implant. One animal (1%) was excluded due to a lost implant. From MRI measurements, 6 out of 28 sessions (21%) needed to be excluded from analyses due to an excessive movement that prevented reliable fMRI analysis. Moreover, 4 sessions (14%) were excluded due to lack of seizures, and 1 session (4%) was excluded due to technical failure in providing air puffs to whiskers. Respiration of awake animals remained stable during the measurements. Average respiration

frequency across the 45 min scans was 2.2±0.4 Hz in visual stimulation group and 2.0±0.3 Hz in whisker stimulation group and no sudden changes of respiration due to stimulations were noted.

## Response to sensory stimuli

Stimulations were manually initiated during ictal and interictal periods, but some stimulation blocks co-occurred unintentionally in-between ictal and interictal periods. *Table 1* shows the amount of each stimulation type and the amount and duration of seizure periods. Regarding statistical power, considering a risk alpha of 0.05, a power of 0.8, matched pairs (seizure/control), we can detect an effect size of 0.37 with 4 animals, considering repeated measurements (4 sessions/animal × 11 seizure/control pairs per session).

Statistical activation maps (*Figure 3*) in response to stimulus were created in interictal and ictal periods and compared between these two states. During interictal condition in the visual stimulation group, statistical responses (p<0.05, cluster-level corrected) were most notably seen in the visual cortex, the superior colliculus, the thalamus (including the lateral geniculate nucleus), and the frontal cortex (including the prelimbic, cingulate, and secondary motor cortices). However, during a seizure condition, responses in the visual cortex were less pronounced while responses in the superior colliculus remained stable. There were more voxels with significant changes of activity during interictal state compared to ictal state (136% more). Comparing the statistical responses between interictal and ictal states revealed significant changes (p<0.05, cluster-level corrected) in the visual, somatosensory, and medial frontal cortices. In the ictal state, these regions were showed significant hemodynamic decreases when comparing to interictal state, and these polarity changes can be seen in the hemodynamic response functions (HRFs) (*Figure 4*).

During the interictal condition in the whisker stimulation group, responses (p<0.05, cluster-level corrected) were most notably seen in the somatosensory cortex, including the barrel field, the auditory cortex, the thalamus, including the ventral posteromedial nucleus, and the medial frontal cortex. During a seizure, responses due to stimulations were less pronounced in the thalamus and the frontal cortex and still present in the somatosensory cortex. There were more voxels with significant changes of activity during interictal state compared to ictal state (179% more). When comparing statistical responses between both states, significant changes (p<0.05, cluster-level corrected) were noticed in the somatosensory, auditory, and frontal cortices: these regions showed significant hemodynamic decreases in ictal state compared to interictal state (see also *Figure 4*).

## Analyses of brain HRFs to sensory stimulations during ictal and interictal brain state

In the visual stimulation group, extreme beta-value of the response in the visual cortex ROI was 2.8±1.7 during the baseline period, and –6.0±2.0 during the seizure period, with a significant difference between the two conditions (p<0.001). Additionally, the response amplitude was higher when the stimulation ended a seizure compared to when it did not (8.1±7.0 to –6.0±2.0, p<0.001). In the whisker stimulation group, the extreme beta-value of the response in the barrel cortex was 4.1±1.9 during the baseline period, and –9.0±1.9 during a seizure, also with a significant difference between the two conditions (p<0.001). In this group, stimulation responses were also higher when the stimulation ended a seizure compared to when it did not (4.8±2.9 to –9.0±1.9, p<0.001). HRFs amplitudes were both negatively and positively signed during the ictal state, depending on the brain region.

When analyzing the effect of a seizure itself, the sessions from the visual and whisker stimulation experiments were pooled together. Significant changes (p<0.05, cluster-level corrected) were observed in various cortical areas, including the frontal, parietal, and occipital regions (*Figure 5A*) totaling 895 voxels. Response amplitudes were predominantly negatively signed in the cortical regions, while they were positively signed in deeper brain regions, such as thalamus and basal ganglia (*Figure 5B*).

## Simulations of sensory stimulation during ictal and interictal periods

*Figure 6* shows the results of the simulations. First, neuronal activity is obtained from the adaptive exponential LIF neurons (AdEx) mean-field for the interictal (*Figure 6A*) and ictal periods (*Figure 6B*). For a better description, the local field potential (LFP) and the membrane potential, calculated from the mean-field model during the SWD type of dynamics, are shown in *Figure 6C*. The LFP was computed

**Table 1.** Characteristics of the stimulations and seizures during a 45 min functional magnetic resonance imaging (fMRI) scanning period. Occurrences (numbers/45 min) of each stimulation type during scanning period are presented, as well as occurrences and duration of seizures.

| | Visual stimulation group | | Whisker stimulation group | |
|---|---|---|---|---|
| **Stimulations (nr/45 min)** | Mean | SD | Mean | SD |
| Stimulation during baseline | 32.4 | 12.3 | 22.2 | 8.8 |
| Stimulation fully inside seizure | 11.4 | 8.2 | 9.9 | 9.5 |
| Stimulation started during seizure, >50% inside of seizure | 2.9 | 3.0 | 2.1 | 2.2 |
| Stimulation started during seizure, >50% outside of seizure | 3.9 | 2.5 | 3.5 | 4.0 |
| Stimulation started before seizure, >50% inside of seizure | 1.5 | 1.6 | 0.1 | 0.3 |
| Stimulation started before seizure, >50% outside of seizure | 1.3 | 1.5 | 0.5 | 0.8 |
| Stimulation ended seizure | 3.1 | 2.8 | 12 | 8.1 |
| Stimulation right after seizure | 1.1 | 0.8 | 1.8 | 1.6 |
| **Seizures** | | | | |
| Total number of seizures | 40.2 | 28.3 | 49.6 | 34.6 |
| Seizures without a stimulation | 15 | 7.8 | 19.8 | 8.2 |
| Duration of seizures (s) | 6.1 | 5.2 | 5.1 | 4.0 |

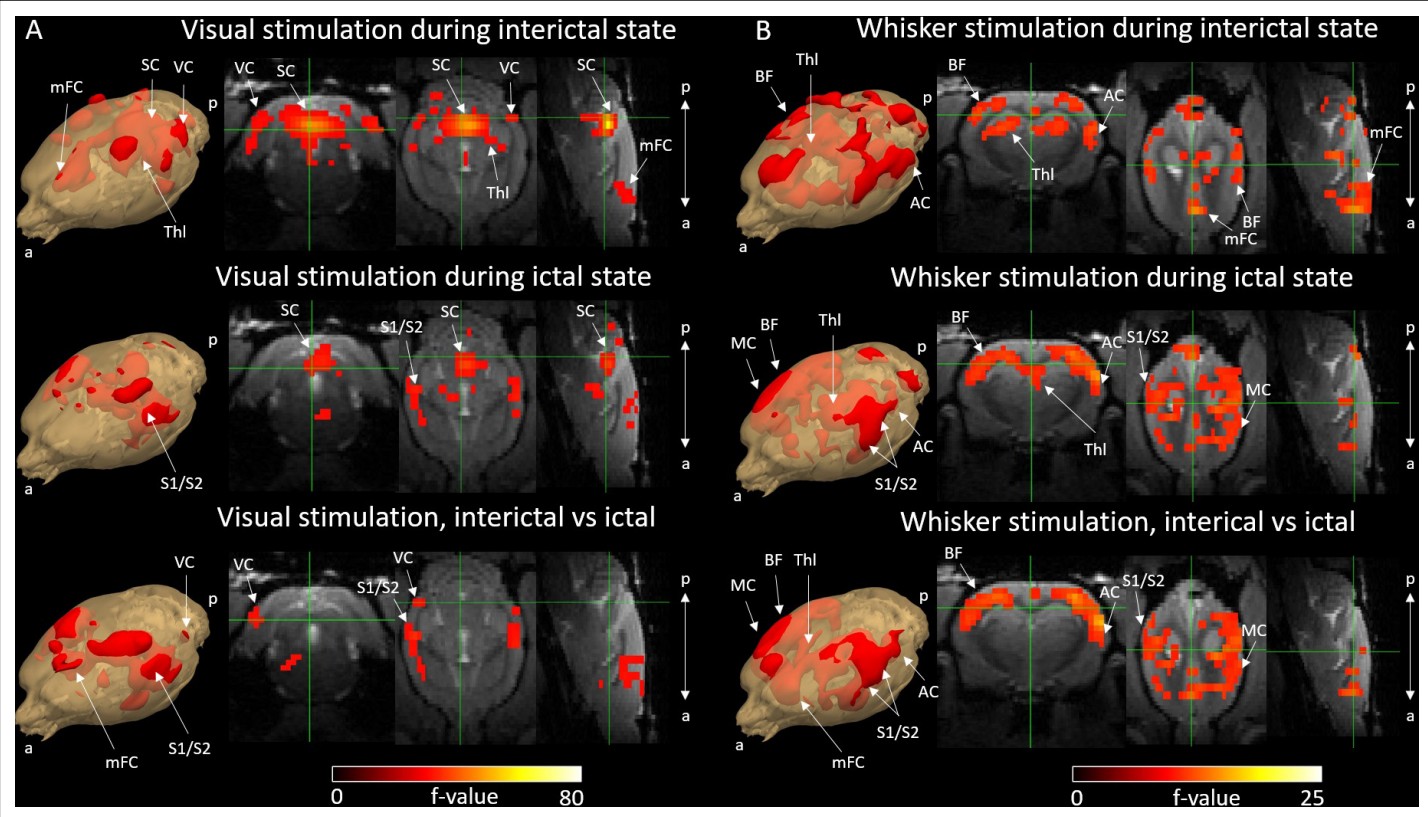

**Figure 3.** Activation F-contrast maps of stimulation responses during interictal and ictal states and difference maps between these two states for the visual (**A**) and the whisker (**B**) stimulation experiments. Parameter estimates of regressors were calculated for every voxel, and contrasts were added to parameter estimates of interictal stimulation, ictal stimulation, and to compare interictal versus ictal stimulation in visual (**A**) and whisker (**B**) stimulation groups. For statistical significance, F-contrast (p<0.05) maps were created and corrected for multiple comparisons by cluster-level correction. AC = auditory cortex, BF = barrel field, mFC = medial frontal cortex, SC = superior colliculus, S1/S2 = primary and secondary somatosensory cortex, Thl = thalamus, VC = visual cortex. a = anterior, p = posterior.

using a recently developed kernel method (*Tesler et al., 2022*). We see that the model can capture an SWD type of pattern, similar to the one observed experimentally in electrophysiological measurements, which is correlated with periods of hyper-polarization in the membrane potential (*Figure 6*, *Figure 6—figure supplement 2*). In the model the hyper-polarization is driven by a strong adaptation current and the SWD dynamics is suppressed by reducing the strength of this current. Thus, the switch between asynchronous irregular (AI) and SWD dynamics in our model is given by varying the strength of the adaptation current.

The time-series and statistical maps of the whole-brain simulations in response to stimulus, performed with the The Virtual Brain (TVB) platform (see Materials and methods), are described in *Figure 6D and E*. The statistical maps for the simulations are calculated directly from the neuronal activity (firing rates). The stimulation of a specific region is simulated as an increase in the excitatory input to the specific node. In particular we use a periodic square function for representing the stimulus (see panel A in *Figure 6—figure supplement 1*). For the results presented here, the stimulus was simulated in the primary visual cortex (indicated by the red-circled pixel). As we can see from the statistical maps, the propagation of the stimulus is drastically restrained during ictal periods in comparison with interictal periods. During the ictal periods the dynamics of the system is dominated by the highly synchronous and regular SWD oscillation, and the effect of the stimulus does not alter significantly the ongoing dynamics. On the contrary, during the interictal periods, where the system exhibits an AI dynamic, the effect of stimulus generates a large variation of the ongoing dynamics in the regions linked to the stimulated area (see also *Figure 6—figure supplement 1*). This agrees with the reduced responsiveness observed in the fMRI results for stimulation during ictal periods.

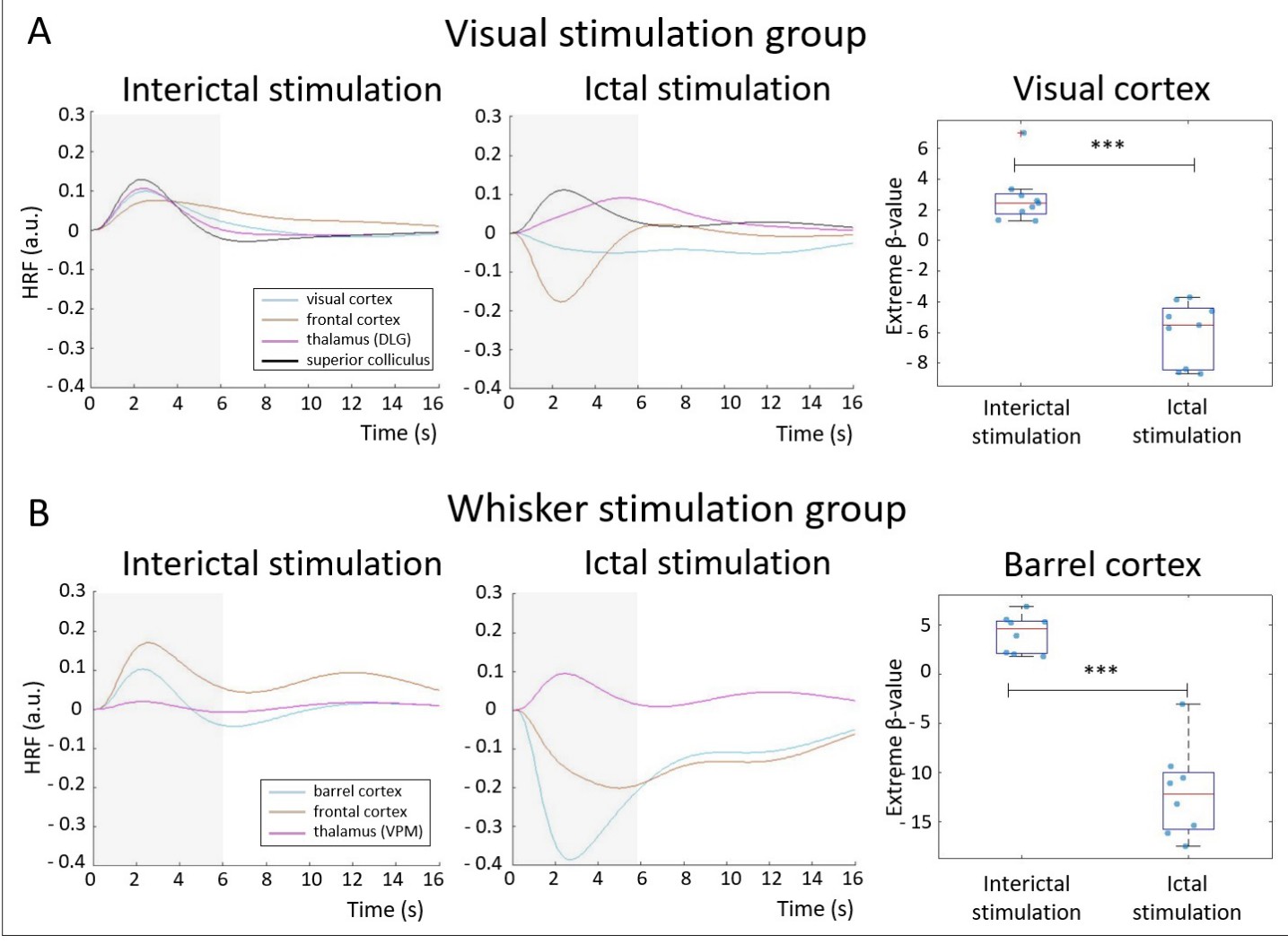

**Figure 4.** Hemodynamic response functions (HRFs) to stimulations performed during an interictal and ictal period: visual stimulation (**A**) and whisker stimulation (**B**) groups. HRFs were calculated in selected ROI, belonging to visual or somatosensory area, by multiplying gamma basis functions (*Figure 1—figure supplement 1B*) with its corresponding average beta-value over a ROI and taking a sum of these values. For statistical comparison, extreme beta-values over a ROI were calculated and values between two states were compared with a two-sample t-test (n=9 sesions in visual stimulation group, n=8 in whisker stimulation group). Scatter plot represents mean ± SD and each blue dot corresponds to extreme beta-values observed from individual functional magnetic resonance imaging (fMRI) sessions. ***=p<0.001. Gray box illustrates stimulation period.

The online version of this article includes the following figure supplement(s) for figure 4:

**Figure supplement 1.** Hemodynamic response functions (HRFs) to stimulation during an ictal period and during a condition when stimulation ended a seizure, for visual stimulation (**A**) and whisker stimulation (**B**) groups.

## Discussion

The investigation of how sensory stimulations are handled by the brain in case of absence epilepsy is essential to develop better care for individuals with this pathology and associated comorbidities. Thanks to specific fMRI and EEG recordings obtained in awake rats, we have uncovered significant differences in brain activity and activation of specific structures during visual and whisker stimulations, depending on whether they occur during a seizure period or not.

### ZTE fMRI of non-curarized awake rats enabled absence seizure detection

Recording absence seizures in rats can be challenging because they only occur when the animals are in a calm and stress-free awake state. Despite the disturbance caused by MRI scanning noise and the

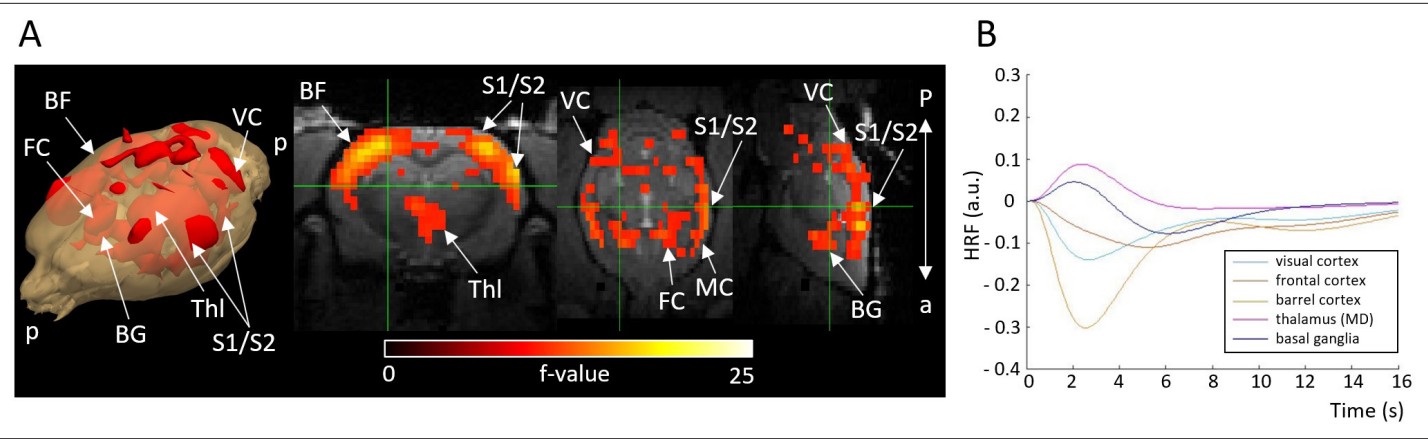

**Figure 5.** Activation F-contrast map (**A**) and hemodynamic response functions (HRFs) to seizure (**B**). Parameter estimates of regressors were calculated for every voxel, and contrasts were added to parameter estimates of seizure in absence of stimulation. For statistical significance, F-contrast (p<0.05) maps were created and corrected for multiple comparisons by cluster-level correction. HRF was calculated in selected ROI by multiplying gamma basis functions (**Figure 1—figure supplement 1B**) with their corresponding average beta-values over a ROI and taking a sum of these values. For both the maps and HRFs, data from visual and whisker stimulation experiments were pooled together. BF = barrel field, BG = basal ganglia, FC = frontal cortex, MD = mediodorsal thalamic nucleus, VC = visual cortex. a = anterior, p = posterior.

need for rat restraint, we were able to successfully detect absence seizures in awake GAERS during training periods and fMRI sessions. We found that the ZTE sequence, which produced considerably lower sound pressures compared to traditional EPI, was the optimal MRI sequence for our purpose. Based on our experience from current and previous experiments, we believe that acoustic noise level is the most significant stress factor for awake rats because it can cause increased motion and thereby potential confounds in task-based and resting-state fMRI studies. ZTE-based fMRI may provide better localization of activated sites because it is sensitive to change in blood flow (**Luh et al., 2000**; **Restom et al., 2007**), and it has also been suggested that ZTE-fMRI is 67% more sensitive than standard BOLD EPI due to its ability to detect increases in tissue oxygenation which shortens the T1-relaxation rate of spins in a pseudo-steady state.

## Whole-brain responsiveness during an interictal condition

In the visual stimulation group, responses were most pronounced in the visual cortex, the superior colliculus, and the thalamus, including the lateral geniculate nucleus, which are all part of the rat visual pathways (**Sefton et al., 2015**). As binocular stimulation was used, responses in both hemispheres were observed. Interestingly, we also detected activation in the frontal cortex, including the prelimbic, cingulate, and secondary motor cortices. The involvement of frontal cortical areas can be explained by the fact that medial prefrontal cortex, which includes the prelimbic cortex, is known to be a central area for rat's attention (**Williams et al., 1999**) and that lesion in the medial prefrontal cortex impaired the accuracy of detecting brief flashes of light (**Muir et al., 1996**).

In the whisker stimulation group, responses were mostly seen in the somatosensory cortex, including the barrel field, and in the ventral posteromedial thalamus, which are part of the rat whisker system (**Adibi, 2019**). This finding suggests that the somatosensory cortex, the initiating zone of the seizures, is functional during the interictal state in GAERS. Additionally, we detected activation in the frontal cortex, which could be due to increased attention to external cues. Surprisingly, positive activation was also observed in part of the auditory cortex. This activation could be explained by the rats' ability to differentiate the sound coming from air puffs from the ongoing MRI noises, causing increased auditory activation during stimulation.

## Decreased cortical activity during a seizure per se

Experimental and simulation results (**Figures 1 and 6**) illustrated typical synchronous and regular SWD patterns with a spike component followed by a longer-lasting wave or silence component. The silence component of SWD is thought to be caused by neuronal deactivation or increased inhibitory activity (**Fisher and Prince, 1977**; **Gloor, 1978**; **Inoue et al., 1993**; **Kostopoulos et al., 1982**;

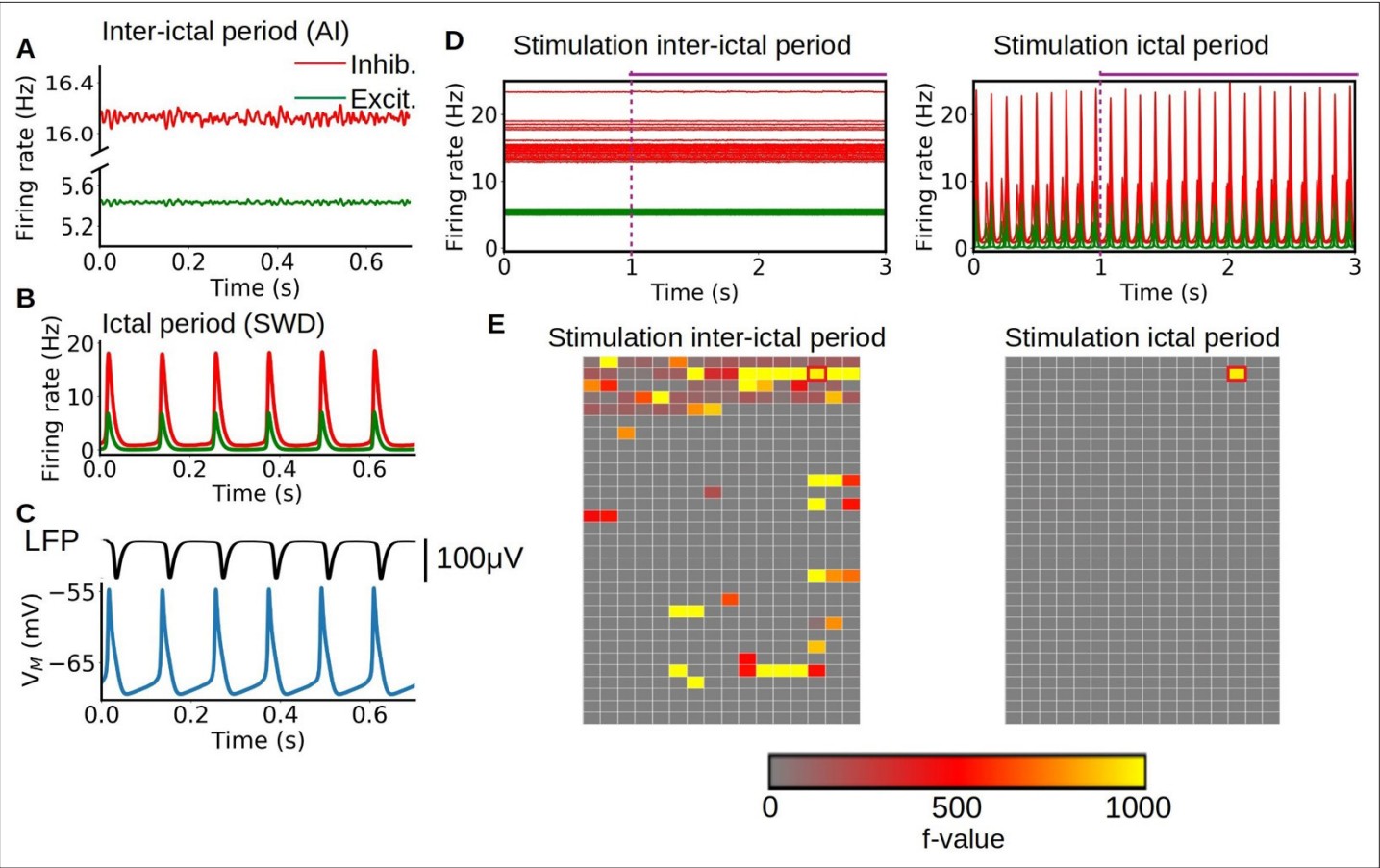

**Figure 6.** Simulation of sensory stimulation during ictal and interictal periods. (A–B) Asynchronous irregular (AI) and spike-and-wave discharge (SWD) type of dynamics obtained from the mean-field model, representing interictal and ictal periods respectively. The change between the two dynamics is given by the strength of the adaptation current in the adaptive exponential LIF neurons (AdEx) mean-field model. (C) Local field potential (LFP) and membrane potential obtained from the mean-field model. The model can capture the SWD pattern observed experimentally in LFP measurements which is correlated with periods of hyper-polarization in the membrane potential. (D–E) Time-series and statistical maps of the simulated sensory stimulus in the whole-brain simulations of the rat, showing the results of a stimulation of the primary visual cortex during ictal and interictal periods. The onset and duration of the stimulus is indicated by the dashed vertical line and horizontal line at the top of the time-series. The statistical maps are built from a 2D representation of the 496 brain regions of the BAMS rat connectome described in the method section "Modeling and simulations".

The online version of this article includes the following figure supplement(s) for figure 6:

**Figure supplement 1.** Single region activity during stimulation in whole-brain simulation.

**Figure supplement 2.** Comparison of the spike-and-wave discharge (SWD) dynamics in the mean-field model and a spiking-neural network of adaptive exponential LIF neurons (AdEx) neurons.

*McCafferty et al., 2023*), potentially resulting in overall decreased neuronal activity in the thalamo-cortical and cortical neurons. Our findings of negative HRFs in the cortical regions (*Figure 5*) are in line with this hypothesis, suggesting reduced neuronal activity. Supporting evidence was found in a recent neuroimaging study conducted with GAERS showing that during an absence seizure, a larger group of neural populations in the frontoparietal cortex had a decreased firing rate, while a smaller portion of neuronal populations had an increased firing rate, leading to reduced fMRI cortical activity (*McCafferty et al., 2023*). Moreover, other studies utilizing optical flowmetry in GAERS have noticed decreased cerebral blood flow (CBF) in cortical capillaries during an absence seizure, with preserved $P_{O_2}$, $P_{CO_2}$, and arterial blood pressure (*Nehlig et al., 1996*) suggesting decreased cortical brain activity. However, the mechanism for the observed subcortical fMRI increases in this study and a previous study (*McCafferty et al., 2023*) illustrating the disagreement between electrophysiological and fMRI signals are yet to explained and demand further studies.

## Decreased responsiveness during an absence seizure

Based on fMRI results, we noticed reduced activation in the cortex during ictal state, along with limited propagation of activity compared to the interictal period (*Figure 4*), which could indicate decreased responsiveness and information processing during external stimulation. Simulation results using a mean-field model also illustrated no observable change in spiking rate together with restricted propagation of neuronal activity in response to visual stimulation (*Figure 6D and E*, *Figure 6—figure supplement 1*), confirming fMRI finding. Previous electrophysiological experimental studies using whisker stimulation setups on GAERS have demonstrated that event-related potentials are modulated but still present during an absence seizure compared to seizure-free periods (*Chipaux et al., 2013*; *Studer et al., 2019*). Therefore, our findings do not necessarily correspond to a decreased amplitude of event-related potential per se but could rather indicate decreased sum of excitatory-inhibitory neuronal state when stimulating during SWD.

During an SWD, the overall neuronal activity in several cortical areas is globally decreased (*McCafferty et al., 2023*) potentially due to the wave component of SWD with increased thalamocortical inhibition (*McCafferty et al., 2018*). In that state, excitation caused by external stimulation can cause relative increases of activity in responsive brain areas. However, if excitation in these neuronal populations is relatively lower compared to the neuronal suppression caused by SWD, the total sum of activity is negative compared to baseline. As fMRI is believed to measure the sum of neuronal activation (*Logothetis et al., 2001*), this appears as reduced fMRI amplitude, which was also apparent in measured HRFs in cortical regions (*Figure 4*). Thus, reduced fMRI responsiveness implies that SWD is a dominant brain feature even under stimulation condition, and that particularly the 'wave' phase of the neuronal oscillatory pattern of SWD can prevent responsiveness in GAERS during these conditions.

As previously discussed by *Chipaux et al., 2013*, the lack of conscious perception to paroxysmal stimulation event could be caused by the concomitant appearance of SWD or SWD complex that is shortly (<500 ms in humans) following a stimulation. In our study, as SWD complexes co-occurred with stimulation events, this could hinder the potential conscious perception of visual and somatosensory stimulation. Interestingly, when stimulation events were applied during a seizure but were not followed by SWD complexes (i.e. stimulation ended a seizure), fMRI response amplitudes were higher than when stimulation did not end a seizure. This could mean that the stimulation response exceeded a threshold to shift the brain back to a responsive or conscious state, which was apparent in the fMRI signal as well. The reason for the change in brain state in these stimulation cases is unclear, as stimulation strength remained the same throughout the experiment. However, there is a potential for future experiments and therapeutic interventions to study further how different stimulation schemes affect neuronal populations and cause a shift from ictal to non-ictal state.

In this study, we were able to detect spontaneous absence seizures inside the magnet using awake, non-curarized, rats and a quiet ZTE imaging sequence, and thereby able to study whole-brain sensory perception in ictal and interictal brain states. Rats were stimulated with blue led light and with somatosensory whisker air puffs during both interictal and ictal states. We found that cortical activation was spatially wider and stronger during interictal stimulation compared to ictal stimulation. Moreover, the detected decreases in the cortical HRF when sensory stimulation was applied during SWDs could play a role in decreased sensory perception. Further studies are required to evaluate whether this HRF change is a cause or a consequence of the reduced neuronal response. Altogether, these stimulation-based fMRI experiments, combined with the brain-wide simulation results, suggest that rats with absence epilepsy have hindered and restricted sensory processing during an ongoing seizure, although not fully abolished. These findings may contribute to our understanding of cognitive impairments observed in patients suffering from absence epilepsy and the severe neuropsychiatric comorbidities observed in children. Additionally, the results of this study provide insight into the information processing during SWD and may aid in the development of future therapeutic approaches for absence epilepsy.

## Materials and methods
### Animals

All experiments were approved by the local animal welfare committee (Comité Local GIN, C2EA-04) and complied with EU guidelines (Directive 2010/63/EU). Every precaution was taken to minimize the

number of animals used and stress to animals during experiments. A total of 11 adult 8–12 months of age GAERS rats were used (260±21 g, 6 males, 5 females). After implantation of an EEG lead (see below), rats were individually housed in their cages. Animals were maintained on a 12/12 hr light-dark cycle at room temperature of 22 ± 2°C, humidity of 50–60%. Food (Extrudat, vitamin-fortified, irradiated >25 kGy) and water were available ad libitum.

## EEG implantation

Carbon fiber (WPI Sarasota FL) electrodes were prepared as follows. Fiber leads were cut to small (~30 mm in length) parts and the end of the electrode was exposed from insulating cover to leave a brush-like, ~5 mm length, tip. Rats were anesthetized with isoflurane (induction with 5%, a percentage gradually decreased to 1–2% for the surgery maintenance). The head was then shaved to remove the fur above the scalp, and the rat was positioned in a stereotaxic frame (David Kopf Instruments, Germany). After local lidocaine hydrochloride injections (2 g/100 ml, 0.05 ml/site), the skull was exposed, cleaned with sterile 0.9% saline and hydrogen peroxide, and allowed to dry. Small holes were drilled halfway through the skull and carbon fiber electrodes were laid and glued with cyanoacrylate over the right motor cortex (AP: +2, ML: +2.5 mm) and right primary somatosensory cortex (AP: −2.5, ML: +3 mm) for seizure detection. An electrode working both as a reference and ground was placed on top of the cerebellum (AP: −12, ML: +2 mm). The other end of electrode leads was inserted inside a plastic tube, which was positioned at the center of the skull. A thin layer of cyanoacrylate was applied to cover the skull. On top of the glue, a layer of dental cement (Selectaplus, DeguDent GmbH, Germany) was applied to finish the implant. Following the surgery, rats were individually caged to recover, and the welfare of the animals was closely monitored.

## Animal habituation for awake imaging

A low noise ZTE MRI sequence (see *Wiesinger and Ho, 2022*) was used with awake rats aiming for low-stress and low-motion functional imaging sessions that allows rats to produce spontaneous seizures. For the habituation, peak ZTE scanner noise was measured with an omnidirectional condenser microphone (MT830R, Audio-Technica Limited, Leeds, UK) and Audacity software (version 2.3.0, https://www.audacityteam.org/), similar to *Paasonen et al., 2020*, and reproduced at equivalent sound pressure through a loudspeaker.

To habituate the rats for the fMRI experiments, a procedure based on a previous study was followed (*Stenroos et al., 2018*). Restraint parts compatible with standard Bruker rat bed and suitable for stimulation leads were designed using 123D CAD software (Autodesk, San Rafael, CA, USA) and 3D-printed with Ultimaker 2 (Utrecht, Netherlands) using acrylonitrile butadiene styrene plastic (*Figure 1*). The rats' bodies were restrained with a soft and elastic foamed plastic, hind legs were taped together, and front legs were taped loosely together to the side of the body. Front teeth were secured with a carbon fiber teeth bar, head with a nose cone, and neck and shoulders with a neck and shoulder bars, respectively. Silicone ear plugs were used to minimize experienced noise. Rats were habituated to restraint and MRI ZTE gradient noises by gradually increasing session times from 15 to 60 min per day for 8 days before the first fMRI experiments. The length of habituation period was selected based on pilot experiments to provide low-motion data therefore giving rats a better chance to be in a low-stress state and thus produce absence seizures inside the magnet. Pressure pillow and video camera were used to estimate physiological state, via breathing rate, and motion level, respectively. During the last habituation session, EEG was measured to confirm that the rats produced a sufficient amount of absence seizures (10 or more per session). Total of three to five fMRI experiments were conducted per rat within a 1- to 3-week period. In case rats were re-imaged more than 1 week after the preceding experiment, an additional 2-day habituation period was conducted (*Figure 1— figure supplement 2*). Before and after each habituation session, rats were given a treat of 1% sugar water and/or three chocolate cereals as positive reinforcement.

## EEG-MRI protocol

MRI acquisitions were carried out at 9.4 T (Biospec Avance III HD, Bruker, Ettlingen, Germany; IRMaGe facility) using Paravision 7. In-house-made transmit-receive loop coil with a 22 mm inner diameter was designed to host the EEG lead and to be compatible with the rat restraining cradle (*Figure 2*). 3D-printed sledge was designed to cover the circuit board of the coil and partly stabilize coaxial

cable and the loop. Before connecting carbon fiber electrode leads, the loop was placed around the implant and sledge was fastened to the animal cradle with masking tape. Next, electrode leads were connected to the other end of the EEG cable. The cable was carefully secured on top of 3D-holder, without touching the animal to avoid breathing and motion artifacts.

## fMRI optimization

To select the optimal fMRI sequence for the study design, standard GE-EPI, and ZTE imaging (see section "EEG-MRI acqustion and stimulations parameters" for details) were compared. Peak acoustic noise levels produced by the MRI scanner during the sequence run, level of spatial distortions on MRI images caused by electrodes, gradient switching artifacts on EEG, and functional contrast were evaluated (*Figure 2*, *Figure 2—figure supplement 1*). Motion incidences between this study and previous study using EPI sequence (*Stenroos et al., 2018*) were compared. ZTE was selected over EPI sequence as it was quieter, produced less susceptibility artifacts, had less noise on EEG recordings, and led to less animal motion (see Results section). Low noise level of ZTE is particularly important factor for seizure appearance, as GAERS rats only experience seizures when they are awake and in a calm state. Eventually, the ZTE parameters were adapted to optimize the functional contrast to noise ratio together with the temporal and spatial resolutions, based on previous reports (MacKinnon et al., ISMRM 2021) and our own pilot acquisitions. Note that ZTE is sensitive to change in blood flow (*Lehto et al., 2017*) and not to BOLD contrast as in EPI, and therefore also more direct measure of neuronal activation. To assess the overall data quality and functionality of the transceiver loop coil, spatial and temporal SNRs were assessed in one representative rat (*Figure 2*). Spatial SNR was obtained as the ratio between the mean signal intensity in an area of interest (the cortex) and the standard deviation of the background signal, whereas temporal SNR was obtained voxel-wise by dividing the mean by the standard deviation of the normalized fMRI signal.

## Animal installation

Before awake EEG-fMRI measurements, rats were anesthetized with isoflurane (5% induction, 2% maintenance in 30% $O_2$/70% $N_2$). Next, animals were wrapped and taped for awake imaging (see section "Animal habituation for awake imaging") and the animals were moved to the MRI scanner and restrained to the 3D-printed holder. The respiration rate was measured with a pressure pillow placed under animals and a Biopac amplifier system (Goleta, CA, USA). Temperature was not measured in awake condition to avoid causing any harms to the rectum due to motion. We relied on pilot calibration of the temperature of heated water circulating inside the animal bed to maintain the normal body temperature of ~37°C. Lastly, EEG leads were connected, and animals were pushed to the center of the magnet bore. After routine preparation steps in the MRI console were done, isoflurane was turned off. Once animals woke up from anesthesia, they were left to rest in a quiet bore for 5–15 min so that spontaneous seizures started to emerge, after which functional imaging was started.

## EEG-MRI acquisition and stimulations parameters

Anatomical imaging was conducted with a T1-FLASH sequence (repetition time: 530 ms, echo time: 4 ms, flip angle 18°, bandwidth 39,682 kHz, matrix size 128×128, 51 slices, field-of-view 32×32 mm², spatial resolution 0.25×0.25×0.5 mm³). fMRI was performed with a 3D ZTE sequence (repetition time: 0.971 ms, echo time: 0 ms, flip angle 4°, pulse length 1 μs, bandwidth 150 kHz, oversampling 4, matrix size 60×60×60, field-of-view 30×30×60 mm³, spatial resolution of 0.5×0.5×1 mm³, polar under sampling factor 5.64, number of projections 2060 resulting to a volume acquisition time of about 2 s) (look *Wiesinger and Ho, 2022* for parameter explanations). A total of 1350 volumes (45 min) were acquired.

Visual (n=14 sessions, 5 rats) and somatosensory whisker (n=14 sessions, 4 rats) stimulations fMRI measurements were performed (*Figure 1A*). Stimulus duration, frequency, and pulse lengths were automatically controlled by an Arduino chip (Arduino Uno Rev3). For the visual stimulation, light pulses (3 Hz, 6 s total length, pulse length 166 ms) were produced by a blue led (wavelength of 470 nm) and light was guided through two optical fibers to the front of the rat's eyes.

For the somatosensory stimulation, an air pressure valve (PMI-200 pressure micro-injector, Dagan Corporation, Minneapolis, MN, USA) controlled by the Arduino chip produced air flow pulses (2 Hz, 6 s total length, pulse length 250 ms) with a pressure of 2.3 bar. Air flow was guided through polyethylene

tubes, ending with a plastic tip, in front of the rat's whiskers, which were maintained together with a piece of tape.

Stimulation parameters were based on previous rat stimulation fMRI studies and chosen to activate voxels widely in visual and somatosensory pathways, correspondingly (*Lu et al., 2016*; *Van Camp et al., 2006*). Both sets of stimulations were initiated manually using the live EEG recording as a guide, either during ictal or interictal state, to pursue equal sampling of both states across each 45 min fMRI session. A delay of at least 20 s was maintained between each stimulation block, to allow hemodynamic responses to settle to baseline between each block. EEG was recorded with a sampling rate of 1024 Hz (Micromed, SD MRI amplifier, Treviso, Italy). EEG recording software received a TTL trigger from the MRI scanner to mark the MRI sequence onset and TTL triggers from the stimulator to mark each stimulation period onset. During the live monitoring, a 50 Hz notch filter, and a low pass filter of 12 Hz facilitated online visual detection of seizures.

## Data analysis

### Motion analysis during fMRI

Motion occurrences were analyzed by visually inspecting image volumes through each scan and average motions per minute were calculated. Motion correction parameters given by advanced normalizing tools (ANTs) (see section "fMRI analysis") were also used and maximum value was taken to estimate maximum displacement of the head from each session. Framewise displacement (*Figure 2E*) was calculated as follows. First, the differential of successive motion parameters (x, y, z translation, roll, pitch, yaw rotation) was calculated. Then absolute value was taken from each parameter, and rotational parameters were divided by 5 mm (as estimate of the rat brain radius) to convert degrees to millimeters (*Power et al., 2012*). Lastly all the parameters were summed together.

### EEG analysis

EEG data was converted from native Micromed TRC-file format to mat- and dat-files using statistical parametric mapping (SPM), version 12 (https://www.fil.ion.ucl.ac.uk/spm/software/spm12/, https://www.fil.ion.ucl.ac.uk/spm/doc/spm12_manual.pdf) toolbox ImaGIN (https://github.com/manikbh/ImaGIN2, copy archived at *manikbh, 2022*). Data were filtered with a 50 Hz notch and 1–90 Hz Butterworth band pass filters using SPM and ImaGIN. Absence seizures were manually inspected from a filtered signal and spectrogram. Seizures were confirmed as SWDs if they had a typical regular spike-and-wave pattern with 7–12 Hz frequency range and had at least double the amplitude compared to baseline signal. All other signals were classified as baseline, i.e., signal absent of a distinctive 7–12 Hz frequency power but spread within frequencies from 1 to 90 Hz. Two successive seizures were counted as one if there was less than 1 s of baseline signal between them. Seizure was counted as absence seizure only if it lasted at least 2 s, since behavioral deficits are not obvious in shorter seizures (*Blumenfeld, 2012*). Seizure initiation and ending time points as well as stimulation onsets were marked in seconds and converted to MRI time in volumes.

Temporal registration between EEG and fMRI was performed using TTL triggers delivered by the MRI sequence and registered as the same time as fMRI signals.

### fMRI analysis

To prepare the fMRI data for analysis, data were motion corrected, co-registered with anatomical MRI, and spatially smoothed to reduce noise level. The preprocessing steps were performed using ANTs (http://stnava.github.io/ANTs/; *Avants et al., 2024*) and Python-based graphical user interphase mri_works (https://montigno.github.io/mri_works/Home/index.html, version 20.08.21a). The first 5 volumes from each fMRI data were removed to allow signal to reach steady state. For motion correction, all volumes were transformed to an average image taken from first 10 volumes by using antsMotionCorr. Normalization was performed indirectly and using the T1-FLASH images, as ZTE images lack anatomical contrast. First, the T1-FLASH images of each animal were corrected from intensity non-uniformity using N4BiasFieldCorrection from ANTs. Next, brain was extracted with MP3 software (*Brossard et al., 2020*). T1-FLASH images of animals were then co-registered on top of reference T1-FLASH image (taken from a representative animal from our study) with ANTs using rigid, affine (linear) and SYN (non-linear) registrations. Transformation matrices were applied to ZTE images

by ANTsApplyTransform so that ZTE images were eventually aligned with the reference T1-FLASH image. Finally, ZTE images were smoothed with a Gaussian filter (1 mm full width at half maximum).

Post hoc, stimulation periods were classified into different inputs (*Figure 1B*, *Figure 1—figure supplement 1A*), based on the relative position in time of the stimulation period with respect to the absence seizure. Stimulation fully applied during an ictal period and those during an interictal period were used for the ictal-interictal comparison analysis. The effects of the periods when stimulation was ending a seizure (stimulation was considered to end a seizure when there was 0–2 s between stimulation start and seizure end) and the effect of the seizures itself, in absence of stimulation, were also studied. The periods when stimulation ended a seizure are particularly interesting for studying the spatial and temporal aspects explaining shift from ictal, to interictal, i.e., potential non-responsiveness state, to responsiveness state. Intermediate cases, where the seizure started or ended during the stimulation block (*Figure 1—figure supplement 1A*), were considered as confounds of no-interest in the SPM general linear model analysis of fMRI data and the explained variance caused by the confounds were reduced from the main effects of interests. Translational and rotational motion parameters, obtained from a motion correction step, were used also as confounds of no-interest. The variance caused by the confounds of no-interests was reduced from the main effects of interests in the linear model. Stimulations that coincided with a motion above 0.3% of the voxel size were not considered as stimulation inputs. To account for temporal and dispersion variations in the hemodynamic response, stimulation and seizure inputs were convolved with three gamma distribution basis functions (i.e. third-order gamma) in SPM (option: basis functions, gamma functions, order: 3). The choice of third-order gamma was based on the expectation that time-to-peak and shape of HRFs of seizure could vary across voxels (*David et al., 2008*).

By convolving each stimulation and seizure inputs with three gamma functions (*Figure 1—figure supplement 1B*), three regressors from each condition were obtained, and those were used to estimate the regression coefficient values (beta-values) for every voxel in the brain. Next, contrasts were applied to the beta-values as a function of the studied stimulation type. For example, to compare the effect of interictal-ictal stimulations, the beta-values corresponding to interictal stimulation were given a contrast of 1, and those corresponding to ictal stimulation were given a contrast of –1. As a result, for each stimulation and seizure condition, three contrast estimates were obtained. From these contrast estimates, statistical F-contrast maps (look for *Friston et al., 1994*) were created and corrected for multiple comparisons by cluster-level correction (*Figures 3 and 5*). F-contrast was used to test differences of any of the three contrast estimates. For difference maps between interictal-ictal stimulations, F-contrast maps were masked with a T-contrast map, where only first-order basis derivative was used, to reveal brain areas where interictal stimulation provided higher amplitude response than ictal stimulation.

HRFs were estimated in ROIs in response to ictal and interictal stimulus (*Figure 4*). Anatomical ROIs, based on Paxinos atlas (Paxinos and Watson rat brain atlas 7th edition), were drawn on the brain areas where statistical differences were seen in activation maps. Also, HRFs were estimated in response to stimulation that ended a seizure and were compared to the response when stimulation did not end a seizure (*Figure 4—figure supplement 1*). Three gamma basis functions, for each condition, were multiplied with their corresponding average beta-values over a selected ROI, resulting in three functions, and the sum of these functions was taken. The shape and time-to-peak of HRFs during an interictal condition were in line with previous literature (*Lambers et al., 2020*) thereby confirming the selection of the third-order gamma basis function to model HRF. For statistical comparison, extreme beta-values (maximum or minimum) in a selected ROI between different conditions were compared using a two-tailed t-test (*Figure 4*).

## Modeling and simulations

Whole-brain simulations of the rat brain during sensory stimulation were performed. Each region of the rat brain was modeled via a recently developed mean-field model of AdEx (*Volo et al., 2019*), which describes the activity of a population of neurons made by excitatory and inhibitory cells (for details on the model, see Sup. Information and *Volo et al., 2019*). This model has been widely tested and used for the simulation of different brain states (*Goldman et al., 2022*), neuronal responsiveness (*Goldman et al., 2022*; *Volo et al., 2019*), and whole-brain dynamics in different species (*Sacha et al., 2024*). In addition, modeling tools to calculate brain signals (such as LFP and BOLD-fMRI) from this

type of mean-fields have been developed (*Tesler et al., 2022*; *Tesler et al., 2023*). The AdEx mean-field model is capable of producing different neuronal dynamics that can be associated with brain activity during interictal and ictal periods (see Results section). The interictal periods were modeled by an AI type of dynamics, while ictal periods were modeled by an oscillatory dynamic which resembles the SWDs observed during absence epilepsy. The mean-field model was combined with a connectivity matrix of the rat brain (BAMS rat connectome, *Bota et al., 2012*) to build a realistic whole-brain simulation of the stimulus propagation during ictal and interictal periods. The simulations were made with TVB platform, which provides a framework to perform large-scale brain simulations (*Sanz-Leon et al., 2015*).

## Acknowledgements

We would like to thank Jan Warnking for giving valuable technical and data analysis assistance. We also thank Argheesh Bhanot for giving us feedback in HRF analysis. This work was supported by the Human Brain Project Third Specific Grant Agreement, project nr. 945539. Grenoble MRI facility IRMaGe is partly funded by the French program Investissement d'avenir run by the Agence Nationale de la Recherche: grant Infrastructure d'avenir en Biologie Santé ANR-11-INBS-0006.

## Additional information

### Funding

| Funder | Grant reference number | Author |
|---|---|---|
| Human Brain Project | Third Specific Grant Agreement project nr. 945539 | Emmanuel L Barbier |
| Agence Nationale de la Recherche | Infrastructure d'avenir en Biologie Santé ANR-11-INBS-0006 | Emmanuel L Barbier |

The funders had no role in study design, data collection and interpretation, or the decision to submit the work for publication.

### Author contributions

Petteri Stenroos, Investigation, Visualization, Methodology, Writing – original draft, Writing – review and editing; Isabelle Guillemain, Conceptualization, Supervision, Investigation, Methodology, Writing – original draft, Project administration, Writing – review and editing; Federico Tesler, Software, Investigation, Writing – original draft, Writing – review and editing; Olivier Montigon, Software, Methodology, Writing – review and editing; Nora Collomb, Vasile Stupar, Methodology, Writing – review and editing; Alain Destexhe, Conceptualization, Supervision, Writing – review and editing; Veronique Coizet, Conceptualization, Supervision, Project administration, Writing – review and editing; Olivier David, Conceptualization, Software, Supervision, Methodology, Project administration, Writing – review and editing; Emmanuel L Barbier, Conceptualization, Data curation, Supervision, Funding acquisition, Investigation, Methodology, Writing – original draft, Project administration, Writing – review and editing, Resources, Validation

### Author ORCIDs

Petteri Stenroos (iD) http://orcid.org/0000-0002-4733-5014
Federico Tesler (iD) http://orcid.org/0000-0001-5093-6913
Alain Destexhe (iD) http://orcid.org/0000-0001-7405-0455
Veronique Coizet (iD) http://orcid.org/0000-0001-5192-6610
Emmanuel L Barbier (iD) https://orcid.org/0000-0002-4952-1240

### Ethics

All experiments were approved by the local animal welfare committee (Comité Local GIN, C2EA-04) and complied with EU guidelines (Directive 2010/63/EU). Every precaution was taken to minimize the number of animals used and stress to animals during experiments.

Reviewer #1 (Public Review): https://doi.org/10.7554/eLife.90318.4.sa1
Reviewer #2 (Public Review): https://doi.org/10.7554/eLife.90318.4.sa2
Reviewer #3 (Public Review): https://doi.org/10.7554/eLife.90318.4.sa3
Author response https://doi.org/10.7554/eLife.90318.4.sa4

## Additional files

### Supplementary files
• MDAR checklist

### Data availability
All raw and preprocessed data, fMRI preprocessing scripts, and scripts used for the simulations are available in Zenodo repository: https://doi.org/10.5281/zenodo.8104455.

The following dataset was generated:

| Author(s) | Year | Dataset title | Dataset URL | Database and Identifier |
|---|---|---|---|---|
| Stenroos P, Guillemain I, Tesler F, Montigon O, Collomb N, Stupar V, Destexhe A, Coizet V, David O, Barbier EL | 2023 | How Absence Seizures Impair Sensory Perception: Insights from Awake fMRI and Simulation Studies in Rats | https://doi.org/10.5281/zenodo.8104455 | Zenodo, 10.5281/zenodo.8104455 |

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

## Appendix 1

### Physiological and methodological considerations

As $P_{O2}$, $P_{CO2}$, and arterial blood pressure were not measured during the fMRI, there is a possibility that they affect the ZTE-fMRI readout, which is sensitive to blood flow changes. However, as animals were awake and given the fine cerebral autoregulation, blood flow values can be expected to be in the normal range. Another concern is whether blood flow remains stable during the seizure, e.g., increases to a level that could hinder stimulus detection. However, previous Doppler flowmetry studies have shown rather decreased than increased blood velocities during an absence seizure in both humans and rats (*Bode, 1992*; *De Simone et al., 1998*; *Nehlig et al., 1996*). Also, our HRF results measured during a seizure in absence of stimulation suggest a similar finding of reduced CBF especially in cortical regions. Even then, we cannot rule out the possibility of CBF changes influencing the results, especially if there are CBF changes caused by non-neuronal origins. We note a caution that presented maps and time courses showing fMRI changes from visual or whisker stimulation during seizures may contain a mixture of both sensory stimulation-related signals and seizure-related signals. To minimize this contamination in the linear model used, we considered both stimulation and seizure-only states as regressors of interest and used seizure-only responses as nuisance regressors to account for error variance. Thereby, the effects caused by the stimulation should be separated as much as possible from the effects caused by the seizure itself.

As the used awake habituation and imaging protocol didn't allow us to avoid the usage of isoflurane during the preparation steps, we cannot rule out the possible effect of using repetitive anesthesia on brain function. However, duration (~15 min) and concentration of anesthesia (~1.5%) during these steps were still moderate, whereas extended durations (1–3 hr) of either single or repetitive isoflurane exposures have been used in previous studies where long-term effects on brain function have been observed (*Long Ii et al., 2016*; *Stenroos et al., 2021*). Moreover, there was a 5–15 min waiting period between the cessation of anesthesia and initiation of fMRI scan, to avoid the potential short-term effects of isoflurane that has been found to be most prominent during the 5 min after isoflurane cessation (*Dvořáková et al., 2022*).

### AdEx mean-field model

The mean-field equations for the AdEx network are given to a first-order by *Volo et al., 2019*:

$$T\frac{dv_{e,i}}{dt} = F_{e,i}\left(W, \bar{v}_e, v_i\right) - v_{e,i}$$
$$\frac{dW}{dt} = -\frac{W}{\tau_w} = bv_e = a\left(\mu_V\left(\bar{v}_e, v_i, W\right) - E_L\right)$$

where $v_{e,i}$ is the mean neuronal firing rate of the excitatory and inhibitory population, respectively, W is the mean value of the adaptation variable, F is the neuron transfer function, i.e., output firing rate of a neuron when receiving excitatory and inhibitory inputs with mean rates $v_e$ and $v_i$ and with a level of adaptation W, a and b are the sub-threshold and spike-triggered adaptation constants, respectively, $t_w$ is the characteristic time of the adaptation variable, T is a characteristic time for neuronal response, $\mu_V$ is the average membrane voltage, and $E_L$ is the leakage reversal potential. The simulations presented in this paper correspond to the parameters: a=0; b=0 for AI state and b=300 pA for SWD dynamics (b=0 for inhibitory neurons); $t_w$ = 200 ms; T=5 ms; $E_L$=–63 mV for excitatory neurons and –65 mV for inhibitory neurons. The main assumption of the model concerns the size of the neuronal population and the characteristic time of the neuronal dynamics. The size of the neuronal populations must be large enough to ensure the validity of a statistical description (starting in the order of thousands of neurons), and the characteristic time of the population dynamics must be slow enough to be captured by the mean-field formalism (with a lower bound in the order of the few milliseconds). These two conditions are satisfied for the system studied in this paper (for further details on the model, see *Volo et al., 2019*).

This model has been widely tested and used for the simulation of different brain states, i.e., asynchronous-irregular vs slow-waves sleep (*Goldman et al., 2022*), neuronal responsiveness (*Goldman et al., 2022*; *Volo et al., 2019*), and whole-brain dynamics (*Goldman et al., 2022*). In addition, modeling tools to calculate brain signals (such as LFP and BOLD-fMRI) from this type of mean-fields have been developed (*Tesler et al., 2022*; *Tesler et al., 2023*). For further validation, we show in *Figure 6—figure supplement 2* the comparison of the SWD type of dynamics in the mean-

field and in the corresponding spiking-neural network of AdEx. We see that, although the amplitude of the oscillations is larger in the spiking network, the mean-field can correctly capture the general dynamics and frequency of the SWD pattern.

## Single region activity during stimulation in whole-brain simulation

For a better illustration of the response to external stimulation during the different states we show in *Figure 6—figure supplement 1* the response of single brain regions during the stimulation protocol for each state. In particular we show the stimulated area (primary visual cortex, V1) and a neighboring region (mediolateral visual area) with strong structural connectivity to V1 (see caption of the figure for details). The stimulation of a specific region is simulated as an increase in the excitatory input to the specific node. In particular, we use a square function for representing the stimulus (see panel A in *Figure 6—figure supplement 1*). As we can see in the figure, during the interictal period (panels A, B, C) the activity is strongly driven by external stimulation and the firing rates follow the stimulation pattern. On the other hand, during the ictal period (panels D, E, F) the activity is mainly driven by the global oscillatory dynamics even during stimulation. We see in panels E, F that the stimulation can lead to a certain variation on the amplitude of the oscillations during some of the cycles, but the neuronal activity remains predominantly dominated by the global ongoing activity (SWD dynamics). This difference in responsiveness between the two states is captured in the statistical maps shown in the main text. To build the statistical maps, an ANOVA (analysis of variance) test was used. This test is originally thought to assess the significance of the change in the mean between two samples and is calculated via an F-test as the ratio of the variance between and within samples. In our case, it allowed us to assess the impact of the stimulation on the ongoing neuronal activity by performing a comparison of the timeseries of the firing rate with and without stimulation (this was performed independently for each state). For the results presented in this paper, the ANOVA was performed using the 'f_oneway' function of the scipy.stats. module in Python.

