## [Editor Report · eLife assessment]

This study conducted fMRI experiments in an inbred rat model of absence seizures. The results provide new information suggesting reduced brain responsiveness during this type of seizure. The reviewers had divergent opinions but on average thought the study was **valuable** and the conclusions were **solid**.

---

## [Referee Report · Reviewer #1 (Public Review)]

In this paper, the effects of two sensory stimuli (visual and somatosensory) on fMRI responsiveness during absence seizures were investigated in GEARS rats with concurrent EEG recordings. SPM analysis of fMRI showed a significant reduction in whole-brain responsiveness during the ictal period compared to the interictal period under both stimuli, and this phenomenon was replicated in a structurally constrained whole-brain computational model of rat brains.

The conclusion of this paper is that whole-brain responsiveness to both sensory stimuli is inhibited and spatially impeded during seizures.

The authors have revised this paper with a lot of detail.

---

## [Referee Report · Reviewer #2 (Public Review)]

Summary:

This study examined the possible effect of spike-wave discharges (SWDs) on the response to visual or somatosensory stimulation using fMRI and EEG. This is a significant topic because SWDs often are called seizures and because there is non-responsiveness at this time, it would be logical that responses to sensory stimulation are reduced. On the other hand, in rodents with SWDs, sensory stimulation (a noise, for example) often terminates the SWD/seizure.

In humans, these periods of SWDs are due to thalamocortical oscillations. A certain percentage of the normal population can have SWDs in response to photic stimulation at specific frequencies. Other individuals develop SWDs without stimulation. They disrupt consciousness. Individuals have an absent look, or "absence", which is called absence epilepsy.

The authors use a rat model to study the responses to stimulation of the visual or somatosensory systems during and in between SWDs. They report that the response to stimulation is reduced during the SWDs. While some data show this nicely, there is also difficulty knowing the effect of the stimulus, SWD and stimulus + SWD.

The authors also study the hemodynamic response function (HRF) and it is not clear what conclusions can be made from the data. The authors acknowledge this, but it does lessen its significance.

Finally, the authors use a model to analyze the data. This model is novel and while that is a strength, its validation is with a second model rather than empirical data.

Strengths:

Use of fMRI and EEG to study SWDs in rats.

Weaknesses:

The paper has been improved by revisions but there are still parts that are unclear, as described below.

---

## [Referee Report · Reviewer #3 (Public Review)]

Summary:

This is an interesting paper investigating fMRI changes during sensory (visual, tactile) stimulation and absence seizures in the GAERS model. The results are potentially important for the field and do suggest that sensory stimulation may not activate brain regions normally during absence seizures. However the findings are limited by substantial methodological issues that do not enable fMRI signals related to absence seizures to be fully disentangled from fMRI signals related to the sensory stimuli.

Strengths:

Investigating fMRI brain responses to sensory stimuli during absence seizures in an animal model is a novel approach with potential to yield important insights.

Use of an awake, habituated model is a valid and potentially powerful approach.

The major difficulty with interpreting the results of this study is that the duration of the visual and tactile stimuli were 6 seconds, which is very close to the mean seizure duration per Table 1. Therefore the HRF model looking at fMRI responses to visual or auditory stimuli occurring during seizures was simultaneously weighting both seizure activity and the sensory (visual or auditory) stimuli over the same time intervals on average. The resulting maps and time courses claiming to show fMRI changes from visual or auditory stimulation during seizures will therefore in reality contain some mix of both sensory stimulation-related signals and seizure-related signals. The main claim that the sensory stimuli do not elicit the same activations during seizures as they do in the interictal period may still be true. However the attempts to localize these differences in space or time will be contaminated by the seizure related signals.

In their repeated responses to this comment the authors have stated that some seizures had longer than average duration, and that they have attempted to model the effects of both seizures and sensory stimulation. However these factors do not mitigate the concern because the mean duration of seizures and sensory stimulation remain nearly identical, and the models used therefore will not be able to effectively separate signals related to seizures and related to sensory stimulation. Hemodynamic models can never in reality represent underlying signals in an orthogonal manner, and are only indirectly related to neural activity.

The only way to truly address the important weakness of this study would be to repeat the experiments using stimulus durations that do not match mean seizure duration, e.g. with much shorter duration stimuli.

The authors have clarified and improved the figure images and their description in the text based on previous specific comments. However, the main weakness in the results remains as summarized above.

Minor comments:

Aside from the concerns listed as weaknesses above which were not addressed, most of the more minor comments were addressed by the authors in the resubmissions. However, the comment made twice previously regarding Figure 6-figure supplement 1 was not addressed. It remains impossible to see any firing rate changes elicited by sensory stimuli during the ictal period in parts E and F of the figure vs. parts B and C (interictal), due to the very different scales used. The seizure signals should be removed or accounted for by the model so that any possible sensory stimulus-related signals could be seen, and/or displayed on the same scale as firing rates without seizures. The authors have simply restated their opinion that it is better to include the SWD dynamics in these figures parts, however this makes the figure wholly unconvincing. It is also concerning that part D (ictal), which is in fact shown on the same scale as part A (interictal), actually shows larger firing rates for both excitatory and inhibitory neurons in visual cortex for sensory stimulation during seizures. This contradicts the claims in the rest of the paper that neural activity and fMRI signals are smaller or are even decreased in visual cortex with sensory stimulation during seizures compared to the interictal period.

---

## [Author Response]

The following is the authors’ response to the previous reviews.

**eLife assessment**
This valuable work performed fMRI experiments in a rodent model of absence seizures. The results provide new information regarding the brain's responsiveness to environmental stimuli during absence seizures. The authors suggest reduced responsiveness occurs during this type of seizure, and the evidence leading to the conclusion is solid, although reviewers had divergent opinions.
**Public Reviews:**

**Reviewer #1 (Public Review):**
In this paper, the effects of two sensory stimuli (visual and somatosensory) on fMRI responsiveness during absence seizures were investigated in GEARS rats with concurrent EEG recordings. SPM analysis of fMRI showed a significant reduction in whole-brain responsiveness during the ictal period compared to the interictal period under both stimuli, and this phenomenon was replicated in a structurally constrained whole-brain computational model of rat brains.The conclusion of this paper is that whole-brain responsiveness to both sensory stimuli is inhibited and spatially impeded during seizures.
**Reviewer #2 (Public Review):**
Summary:This study examined the possible affect of spike-wave discharges (SWDs) on the response to visual or somatosensory stimulation using fMRI and EEG. This is a significant topic because SWDs often are called seizures and because there is non-responsiveness at this time, it would be logical that responses to sensory stimulation are reduced. On the other hand, in rodents with SWDs, sensory stimulation (a noise, for example) often terminates the SWD/seizure.In humans, these periods of SWDs are due to thalamocortical oscillations. A certain percentage of the normal population can have SWDs in response to photic stimulation at specific frequencies. Other individuals develop SWDs without stimulation. They disrupt consciousness. Individuals have an absent look, or "absence", which is called absence epilepsy.The authors use a rat model to study the responses to stimulation of the visual or somatosensory systems during and in between SWDs. They report that the response to stimulation is reduced during the SWDs. While some data show this nicely, the authors also report on lines 396-8 "When comparing statistical responses between both states, significant changes (p<0.05, cluster-) were noticed in somatosensory auditory frontal..., with these regions being less activated in interictal state (see also Figure 4). That statement is at odds with their conclusion. I do not see that this issue was addressed.

See comments below starting with “We acknowledge the reviewer…”.

They also conclude that stimulation slows the pathways activated by the stimulus. I do not see any data proving this. It would require repeated assessments of the pathways in time. This issue was not addressed.

See comments below starting with “We acknowledge the reviewer…”.

The authors also study the hemodynamic response function (HRF) and it is not clear what conclusions can be made from the data. This is still an issue. No conclusions appear to be possible to make.

See comments below starting with “We acknowledge the reviewer…”.

Finally, the authors use a model to analyze the data. This model is novel and while that is a strength, its validation is unclear. The authors did not add any validation of their model.

See comments below starting with “We acknowledge the reviewer…”.

Strengths:Use of fMRI and EEG to study SWDs in rats.Weaknesses:Several aspects of the Methods and Results were improved but some are still are unclear.

We acknowledge the reviewer for the concerns of we not addressing the comments above. However, we emphasize that most of the comments were addressed in the already sent “Response to Review Comments” and in the updated manuscript. Here we repeat the responses and provide also additional clarifications to some of the comments.

We thank the reviewer for noting the discrepancy in the statement of “less activated in interictal state”. The statement should have been written vice versa. We also address that the direction of activation change between groups can be misinterpreted based on statistical maps itself (Figure 3) where only statistical changes are visible and not the polarity of response (can be seen in Figure 4). Therefore, we have made a following changes in the section 3.3.: “There were more voxels with significant changes of activity during interictal state compared to ictal state (136% more). Comparing the statistical responses between interictal and ictal states revealed significant changes (p<0.05, cluster-level corrected) in the visual, somatosensory, and medial frontal cortices. In the ictal state, these regions showed significant hemodynamic decreases when comparing to interictal state, and these polarity changes can be seen the hemodynamic response functions (Figure 4).”

We agree with the reviewer that there are no data showing slowing of the pathways in response to stimulus. However, we are a bit confused about this comment, as to what part in conclusion section it refers to. We did not intentionally claim that stimulation slows the activated pathways in the manuscript.

Reviewer is right that strong claims cannot be made from HRF by itself. Therefore, we have avoided to such phrasing throughout the manuscript. In the conclusion section, we speculate that HRF decreases “could play a role in decreased sensory perception” but also state that “further studies are required”. The observed HRF decreases (rather than increases) in the cortex when stimulation was applied during SWD, was discussed in section 4.4., where we speculated that neuronal suppression (possible apparent in negative HRFs) caused by SWD can prevent responsiveness. Conclusion now states the following: “Moreover, the detected decreases in the cortical HRF when sensory stimulation was applied during spike-and-wave discharges, could play a role in decreased sensory perception. Further studies are required to evaluate whether this HRF change is a cause or a consequence of the reduced neuronal response.”

We point out that the main validation of the model and its details were provided in the previous answer to the reviewer and added to the manuscript. The model presented in the paper is based on a mean-field formalism that captures neuronal activity at the mesoscale level. This mean-field formalism is derived via a detailed statistical description of the activity of a spiking neuronal population of excitatory and inhibitory with conductance-based synaptic interactions. Thus, the validation of the mean-field model is performed via direct comparison between the dynamics obtained from the mean-field model and the dynamics obtained from the underlying spiking neural network model. This comparison is shown in the supplementary material of the manuscript, where the transition studied in the paper between interictal (asynchronous irregular activity) and ictal (SWD dynamics) activity, which is predicted by the mean-field model, is indeed observed in the underlying spiking neuronal model. The existence of these two types of dynamics and the transition between them is the main component of the model used to build the analysis of the responsiveness performed in the paper (which has been properly validated).

**Reviewer #3 (Public Review):**
Summary:This is an interesting paper investigating fMRI changes during sensory (visual, tactile) stimulation and absence seizures in the GAERS model. The results are potentially important for the field and do suggest that sensory stimulation may not activate brain regions normally during absence seizures. But the findings are limited by substantial methodological issues that do not enable fMRI signals related to absence seizures to be fully disentangled from fMRI signals related to the sensory stimuli.Strengths:Investigating fMRI brain responses to sensory stimuli during absence seizures in an animal model is a novel approach with potential to yield important insights.Use of an awake, habituated model is a valid and potentially powerful approach.Weaknesses:The major difficulty with interpreting the results of this study is that the duration of the visual and tactile stimuli were 6 seconds, which is very close to the mean seizure duration per Table 1. Therefore the HRF model looking at fMRI responses to visual or auditory stimuli occurring during seizures was simultaneously weighting both seizure activity and the sensory (visual or auditory) stimuli over the same time intervals on average. The resulting maps and time courses claiming to show fMRI changes from visual or auditory stimulation during seizures will therefore in reality contain some mix of both sensory stimulation-related signals and seizure-related signals. The main claim that the sensory stimuli do not elicit the same activations during seizures as they do in the interictal period may still be true. But the attempts to localize these differences in space or time will be contaminated by the seizure related signals.In their response to this comment the authors state that some seizures had longer than average duration, and that they attempted to model the effects of both seizures and sensory stimulation. However these factors do not mitigate the concern because the mean duration of seizures and sensory stimulation remain nearly identical, and the models used therefore will not be able to effectively separate signals related to seizures and related to sensory stimulation.

Regressors for seizures were formed by including periods of seizures without any stimulation present. In theory, if seizures were perfectly modeled by the regressor, the left variance is completely orthogonal to the main effect of the stimulus. Furthermore, only the cases where the seizures are longer than the stimulus are used to calculate the responsiveness of the stimulus (while the cases where the seizures are shorter than the stimulus are used as nuisance regressors to account for error variance). However, we agree with the reviewer that in practice all effects of the seizure cannot be removed completely from the effect of stimulus. We have addressed this concern in the “physiologic and methodology consideration” section: “We note a caution that presented maps and time courses showing fMRI changes from visual or whisker stimulation during seizures may contain a mixture of both sensory stimulation-related signals and seizure-related signals. To minimize this contamination in the linear model used, we considered both stimulation and seizure-only states as regressors of interest and used seizure-only responses as nuisance regressors to account for error variance. Thereby, the effects caused by the stimulation should be separated as much as possible from the effects caused by the seizure itself.”

The claims that differences were observed for example between visual cortex and superior colliculus signals with visual stim during seizures vs interictal remain unconvincing due to above.Maps shown in Figure 3 do not show clear changes in the areas claimed to be involved.In their response the authors enlarged the cross sections. However there are still discrepancies between the images and the way they are described in the text. For example, in the Results text the authors say that comparing the interictal and ictal states revealed less activation in the somatosensory cortex during the ictal than during the interictal state, yet Figure 3 bottom row left shows greater activation in somatosensory cortex in this contrast.

We note that the direction of activation change between groups can be misinterpreted based on statistical maps itself (Figure 3) where only statistical changes are visible and not the polarity of response (can be seen in Figure 4). Therefore, we have made the following changes to the section 3.3.: “There were more voxels with significant changes of activity during interictal state compared to ictal state (136% more). Comparing the statistical responses between interictal and ictal states revealed significant changes (p<0.05, cluster-level corrected) in the visual, somatosensory, and medial frontal cortices. In the ictal state, these regions showed significant hemodynamic decreases when comparing to interictal state, and these polarity changes can be seen the hemodynamic response functions (Figure 4).”

**Recommendations for the authors:**

**Reviewer #1 (Recommendations For The Authors):**

Authors have revised this paper with a lot of detail. The paper can be accepted for publication in this version.

**Reviewer #2 (Recommendations For The Authors):**
Reviewer #1(1) The analysis in this paper does not directly answer the scientific question posed by the authors, which is to explore the mechanisms of the reduced brain responsiveness to external stimuli during absence seizures (in terms of altered information processing), but merely characterizes the spatial involvement of such reduced responsiveness. The same holds for the use of mean-field modeling, which merely reproduces experimental results without explaining them mechanistically as what the authors have claimed at the head of the paper.We agree with the reviewer that the manuscript does not answer specifically about the mechanisms of reduced brain responsiveness. The main scientific question addressed in the manuscript was to compare whole-brain responsiveness of stimulus between ictal and interictal states. The sentence that can lead to misinterpretations in the manuscript abstract: "The mechanism underlying the reduced responsiveness to external stimulus remains unknown." was therefore modified to the following "The whole-brain spatial and temporal characteristics of reduced responsiveness to external stimulus remains unknown".This change did not address the issue. The problem is that there is no experimentation to address the underlying mechanisms of the results. I also think the changed language in the abstract is less clear than the original.

We fully agree that this manuscript does not answer or claim to be answering about the mechanisms of reduced brain responsiveness. The main scientific question addressed in the manuscript was to compare whole-brain responsiveness of stimulus between ictal and interictal states, by means of hemodynamics and mean-field simulation.

We have changed the language of the abstract to the following:

“In patients suffering absence epilepsy, recurring seizures can significantly decrease their quality of life and lead to yet untreatable comorbidities. Absence seizures are characterized by spike-and-wave discharges on the electroencephalogram associated with a transient alteration of consciousness. However, it is still unknown how the brain responds to external stimuli during and outside of seizures.

This study aimed to investigate responsiveness to visual and somatosensory stimulation in GAERS, a well-established rat model for absence epilepsy. Animals were maintained in a non-curarized awake state allowing for naturally occurring seizures to be produced inside the magnet. They were imaged continuously using a quiet zero-echo-time functional magnetic resonance imaging (fMRI) sequence. Sensory stimulations were applied during interictal and ictal periods. Whole brain responsiveness and hemodynamic responses were compared between these two states. Additionally, a mean-field simulation model was used to mechanistically explain the changes of neural responsiveness to visual stimulation between interictal and ictal states.

Results showed that, during a seizure, whole-brain responses to both sensory stimulations were suppressed and spatially hindered. In several cortical regions, hemodynamic responses were negatively polarized during seizures, despite the application of a stimulus. The simulation experiments also showed restricted propagation of spontaneous activity due to stimulation and so agreed well with fMRI findings. These results suggest that sensory processing observed during an interictal state is hindered or even suppressed by the occurrence of an absence seizure, potentially contributing to decreased responsiveness during this absence epileptic process.”

The authors also study the hemodynamic response function (HRF) and it is not clear what conclusions can be made from the data.The response of the authors did not clarify this issue. Instead, they explained why they examined HRF and that they can only speculate what the data means.

Reviewer is right that strong claims cannot be made from HRF by itself. Therefore, we have avoided to such phrasing throughout the manuscript. In the conclusion section, we speculate that HRF decreases “could play a role in decreased sensory perception” but also state that “further studies are required”.

Finally, the authors use a model to analyze the data. This model is novel and while that is a strength, its validation is unclear. The conclusion is that the modeling supports the conclusions of the study, which is useful.Details about the model were added.This is not entirely satisfactory because there is still no validation of the model.

We point out that the main validation of the model and its details were provided in the previous answer to the reviewer and added to the manuscript. The model presented in the paper is based on a mean-field formalism that captures neuronal activity at the mesoscale level. This mean-field formalism is derived via a detailed statistical description of the activity of a spiking neuronal population of excitatory and inhibitory with conductance-based synaptic interactions. Thus, the validation of the mean-field model is performed via direct comparison between the dynamics obtained from the mean-field model and the dynamics obtained from the underlying spiking neural network model. This comparison is shown in the supplementary material of the manuscript, where the transition studied in the paper between interictal (asynchronous irregular activity) and ictal (SWD dynamics) activity, which is predicted by the mean-field model, is indeed observed in the underlying spiking neuronal model. The existence of these two types of dynamics and the transition between them is the main component of the model used to build the analysis of the responsiveness performed in the paper (which has been properly validated).

How is ROI defined in this paper? What type of atlas is used?

Anatomical ROIs were drawn based on Paxinos and Watson rat brain atlas 7th edition. Region was selected if there were statistically significant activations detected inside that region, based on activation maps. We clarified the definition of ROI as the following:

"Anatomical ROIs, based on Paxinos atlas (Paxinos and Watson rat brain atlas 7th edition), were drawn on the brain areas where statistical differences were seen in activation maps."

This is helpful, but the unstained brain does not show the borders of the areas. Therefore just saying an atlas was used is not enough. How in an unstained brain can the areas be accurately outlined?

Areas of the brain were differentiated by co-registering the functional MRI images with an T1-weighted anatomical reference brain that was created on site from the same data set that was used for the manuscript. Potential co-registration inaccuracies created by using a reference brain measured in different site, sequence and a rat strain can be thus avoided. T1-images create sufficient contrast to differentiate main brain areas, but for more accurate border definition (e.g., to differentiate different thalamic nuclei), a coordinate system of the atlas and coordinates known in the used anatomical brain, were used to pinpoint exact borders of the brain areas.

Reviewer #2The following also is not precise:"Although seizures are initially triggered by hyperactive somatosensory cortical neurons, the majority of neuronal populations are deactivated rather than activated during the seizure, resulting in an overall decrease in neuronal activity during SWD (McCafferty et al. 2023)."What neuronal populations? Cortex? Which neurons in the cortex? Those projecting to the thalamus? What about thalamocortical relay cells? Thalamic gabaergic neurons?Please check that these issues were corrected.

The issues were addressed as follows:

“Although SWDs are initially triggered by hyperactive somatosensory cortical neurons, neuronal firing rates, especially in majority of frontoparietal cortical and thalamocortical relay neurons, are decreased rather than increased during SWD, resulting in an overall decrease in activity in these neuronal populations (McCafferty et al., 2023). Previous fMRI studies have demonstrated blood volume or BOLD signal decreases in several cortical regions including parietal and occipital cortex, but also, quite surprisingly, increases in subcortical regions such as thalamus, medulla and pons (David et al., 2008; McCafferty et al., 2023).”

ResultsAfter removing problematic animals and sessions, was there sufficient power? There probably wasn't enough to determine sex differences.After removing problematic sessions, we found statistically significant results (multiple comparison corrected) results in both activation maps, and hemodynamic responses. To determine sex differences, there were not enough animals for statistical findings (p>0.05).This is not the question. The question is whether there was sufficient power.

A simple power calculation was performed as follows: considering a t-test, a risk alpha of 0.05, a power of 0.8, matched pairs (seizure/control), we can detect an effect size of 0.37 with our 4 animals, considering repeated measurements (4 sessions/animal x 11 seizure/control pairs per session). This is now mentioned in the manuscript.

Table 1 has no statistical comparisons.

Table 1 is purely an illustration of stimulation and seizure occurrence. There is no specific interest to compare stimulation types (in what state of seizure it occurred) as it does not provide any meaningful inferences to the study.

Table 1 could be improved by statistics. More could be said and there would be justification to include it.

We thank the reviewer for the suggestion, but as it is yet unclear to what statistical comparison would be feasible to do, we opt to leave it out.

Statistical activation maps - it is not clear how this was done.Creation of statistical maps are explained in section 2.5.3.This section is not clear.

We have added a reference (https://doi.org/10.1002/hbm.460020402) for readers to familiarize themselves with the concept of statistical parametric mapping.

Fig 3 "F-contrast maps." Please explain.Creation of statistical maps are explained in section 2.5.3.This section is unclear.

We have added a reference (https://doi.org/10.1002/hbm.460020402) for readers to familiarize themself with the concept of statistical parametric mapping.

**Reviewer #3 (Recommendations For The Authors):**
Aside from the concerns listed as weaknesses above which were not addressed, most of the more minor comments were addressed by the authors in the resubmission. However, the comment below was not addressed because it is impossible to see any firing rate changes elicited by sensory stimuli (if they are present) due to the scale during seizures. The seizure signals should be removed or accounted for by the model so that any possible sensory stimulus-related signals could be seen, and displayed on the same scale as firing rates without seizures. Prior comment (unaddressed) is repeated below:Figure 6-figure supplement 1, the scales are very different for many of the plots so they are hard to compare. Especially in the ictal periods (D, E, F) it is hard to see if any changes are happening during ictal stimulation similar to interictal stimulation due to very different scales. The activity related to SWD is so large that it overshadows the rest, and perhaps should be subtracted out.

These two comments were addressed and replied in the previous round of reviews. Regarding the different scales of the plots from Figure 6-figure supplement 1, we point out that all the plots in the same scale are already presented in Figure 6 of the main-text. Regarding the activity related to SWD and sensory stimulation, we remark that the effect of the stimulation should be (and was) evaluated with respect to the ongoing activity. All the results concerning the neuronal responsiveness presented in the paper evaluate the statistical significance of the changes in activity produced by the stimulation with respect to the ongoing activity (during ictal and interictal states respectively). For this reason, all the plots containing the time series of neuronal activity in the simulations include the ongoing activity (with SWD dynamics when present) for proper comparison and relevant analysis.

Additional changes:

In the section 3.2., the sentence: “In addition, responses were observed in the somatosensory cortex during a seizure state.” was removed for clarification purposes as deactivation rather than activation was observed in this brain area during a seizure state.